# Relative age effect? No "flipping" way! Apparatus dependent inverse relative age effects in elite, women's artistic gymnastics

**Eleanor Langham-Walsh**[ID]*, **Victoria Gottwald, James Hardy**[ID]

Institute for the Psychology of Elite Performance, School of Sport, Health and Exercise Sciences, Bangor University, Bangor, Gwynedd, Wales, United Kingdom

* eleanor.langham-walsh@bangor.ac.uk

## Abstract

In contrast to research on team-sports, delayed maturation has been observed in higher-skilled gymnasts, leading to atypical distributions of the relative age effect. Recent studies have reported intra-sport differences in the relative age effect and given the task demands across gymnastics apparatus, we expected to find evidence for the influence of apparatus specialism. We examined the presence of a relative age effects within a sample of elite, international, women's artistic gymnasts ($N$ = 806, $N_{countries}$ = 87), and further sampled our data from vault, bars, beam, and floor major competition finalists. Poisson regression analysis indicated no relative age effect in the full sample ($p$ = .55; $R^2_{adj.}$ = .01) but an effect that manifested when analysing apparatus independently. The Index of Discrimination ($I_D$) analysis provided evidence of an inverse relative age effect identified for beam ($p$ = .01; $I_D$ = 1.27; $R^2_{adj.}$ = .12), a finding that was corroborated by a marginal effect in our vault finalists ($p$ = .08; $I_D$ = 1.21; $R^2_{adj.}$ = .06). These novel findings can be attributed to the integrated influence of self-fulfilling prophecy upon coach and gymnast expectations, as well as the technical mechanisms underpinning skill development involved in the underdog hypothesis.

## Introduction

A consistent finding within the talent identification and development literature is the influence of an athlete's age in relation to their peers [1]. The relative age effect (RAE) [2] is a phenomenon whereby the chronological age-grouping of children and adolescents can lead to an over-representation of athletes born earlier in the year within a cohort. Inherent in sporting and education systems, children and adolescents are frequently grouped together based on chronological age; for example, a child's birth month within the British September to August school year influences which school year they are assigned to [1]. However, within this type of grouping there can be nearly 12 months difference between the oldest and youngest, leading to a variation in cognitive [1], physical [3], and emotional [4] development. One of the more conventional explanations of the RAE in sport is the maturation-selection hypothesis [1], which assumes enhanced anthropometric characteristics as a function of chronological age.

**Data Availability Statement:** The data underlying the results presented in the study are available from https://thegymter.net/gymnast-database/. This is third party data that others will be able to

access in the same manner as the authors. We confirm that we did not have any special access privileges that others would not have.

**Funding:** This research was conducted while the corresponding author, ELW, was studying for a PhD funded by the Economic and Social Research Council (NE/L002604/1) [https://esrc.ukri.org/; grant number available upon acceptance] and UK Sport [https://www.uksport.gov.uk/; Pathway 2 Podium]. The funders had no role in study design, data collection and analysis, decision to publish, or preparation of the manuscript.

**Competing interests:** The authors have declared that no competing interests exist.

These developmental advantages may manifest in a number of ways including: stature and mass; speed [5]; and greater muscular strength and aerobic power [6]. Ultimately, this results in a selection bias towards relatively older athletes, which provides enhanced access to coaching and resources, further exacerbating the effect [7].

Additional purported underpinnings of the RAE include a broader spectrum of multidisciplinary mechanisms. Psychological approaches adopt the notion of self-fulfilling prophecy [8], whereby behaviours grounded on what may be false beliefs can lead to these perceptions coming true; these behaviours can take the form of Pygmalion and/or Galatea effects. Pygmalion effects occur when an athlete is influenced by expectations from others, such as a coach investing more time into an athlete because they display higher levels of physical prowess. In line with this notion, Peña-González et al. [9] found that coaches held greater expectations for soccer players born within the first quarter of the year (Q1) in comparison to those born in the last quarter (Q4). Similarly, Galatea effects can occur when an athlete is influenced by their self-expectations; for example, increasing practice hours as a reflection of their self-beliefs about their high potential [10].

Whilst the above mechanisms support the robust RAE within sport [1], there is emerging evidence of inter-sport differences (between sport differences). For instance, within gymnastics where atypical birth date distributions have been reported, these findings are likely a result of biases towards delayed-maturation for success [11]. More specifically, Hancock et al. [11] report null effects within a sample of female gymnasts. The lack of a RAE remained when their sample was broken down into regional, provincial, elite-provincial, and national competitive standards for the under-15 age group, as well as national competitive standard for the over-15 age group. This same null effect was also identified by Delaš Kalinski et al. [12, 13] in their respective samples of female and male Olympic gymnasts. The authors accounted for this null finding as a consequence of the advantage of later maturation for the relatively younger gymnasts and the advanced cognitive maturity of those that are relatively older cancelling each other out [12]. Whilst there was no RAE in the national standard over-15 age group, when all over-15 standards were combined Hancock et al. [11] found a reversed RAE. The authors attributed this to the biomechanical advantages possessed by relatively younger athletes post puberty where, due to smaller cognitive discrepancies post maturation, relatively older gymnasts could no longer offset this advantage.

The development of theoretically driven hypotheses regarding nuances in the RAE has led researchers to begin to examine intra-sport differences (within sport differences). These intra-sport differences are typically a consequence of variations in task demands dependent on an athlete's role within their sport. For example, Brustio et al. [14] examined the prevalence of RAEs across different track and field disciplines. Whilst there was a consistent RAE favouring relatively older athletes, this effect was stronger within events that are particularly influenced by the anthropometric and strength qualities of athletes (e.g., hurdles and throwing). Similarly, Jones et al. [15] found positional differences in super-elite rugby union players, wherein a Q1 effect was found for the backs (where there was a greater distribution of backs born in the first quarter of the year), yet the reverse, a Q4 effect, was observed for forwards (where there was a greater distribution of forwards born in the last quarter of the year).

Jones and colleagues [15] reasoned that these differences were due to the respective qualities required across the positions. The overrepresentation of Q4 rugby forwards could be attributed to a "rocky road" developmental trajectory (see Collins & MacNamara [16]), whereby challenge promotes the development of resilience and mental toughness needed to succeed at the elite level. Similarly, the 'underdog' hypothesis [17], has been presented in these contexts to account for the paradoxical benefits of challenge experienced by relatively younger athletes competing against their older counterparts.

Compared to rugby, the nature of task demands in gymnastics is equally if not further varied across apparatus and thus, it stands to reason that we expect to see differences in RAE as a function of apparatus specialism. Research investigating the RAE within individual sports, especially gymnastics, is sparse and the examination of apparatus differences is an original and practically relevant development for the literature. The present study examined apparatus-differences for the RAE in international standard, women's artistic gymnastics, a relatively neglected sport and expertise level within the research literature. Our hypotheses were twofold; first, based on previous studies in gymnastics [11, 12], we did not expect to see a RAE within a sample of elite gymnasts when our sampling ignored apparatus specialism. Second, and arguably the more valuable contribution to the knowledge base, we hypothesised a change in RAE dependent on task demands across different gymnastics apparatus (e.g., power requirements necessary for vault versus the levels of agility required for the beam).

## Materials and methods

### Participants

**Full sample of international gymnasts.** Our initial sample of elite, international gymnasts was obtained from "The Gymternet" gymnast database [18] using the rvest package [19] in R Studio [20]. The database contains archival data on women's artistic gymnasts who have competed at major international championships from 2015 to time of writing ($N = 806$, $M_{age} = 20.63$, $N_{countries} = 87$). The sample included gymnasts that were currently competing in junior (U16; $n = 95$, $M_{age} = 15.69$, $n_{countries} = 42$) and senior (n = 493, $M_{age} = 20.66$, $n_{countries} = 76$) age groups. We did not explore a country effect as these results would have been underpowered in relation to our power calculation. For a summary of the representation of each country within the analysis, please see the S1 Table.

**Apparatus specialists.** A separate sample of apparatus specialists was comprised of gymnasts who had made an Olympic, World or European apparatus final from 2006 (where the current scoring system was first adopted) to 2019. Dates of births were obtained through English Wikipedia. Vault ($n = 91$, $M_{age} = 25.14$, $n_{countries} = 30$); Uneven Bars ($n = 93$, $M_{age} = 24.37$, $n_{countries} = 21$); Beam ($n = 117$, $M_{age} = 24.48$, $n_{countries} = 23$); Floor ($n = 105$, $M_{age} = 24.48$, $n_{countries} = 23$).

### Analysis

We adopted an analytical strategy, in line with recent RAE investigations [14, 21], by employing Poisson regression analysis to analyse our data. Poisson regression uses an explanatory variable (x) to explain the rate of an event (y) using the formula $y = e^{b_0 + b_1 x}$. Within our study, x was the week of birth in the January–December year measured as a decimal fraction within a one-year interval (0,1; $T_b$), and y the rate of births per given week. To calculate $T_b$, birth week ($W_b$) of each athlete was transformed using the formula $T_b = (W_b—0.5)/52$ [14, 22] with .5 referring to the midpoint of the week. Doyle and Bottomley [21] recommend that authors do not produce a simplified odds ratio (e.g. comparing Q1 to Q4) as it only explores set intervals and ignores a large range of points. Therefore we calculated the Index of Discrimination ($I_D$) using the formula $e^{-b}$ [21, 22] which provides a standardised relative odds for a gymnast born at the start of the year in comparison to the end of the year that allows comparison across future studies. We also adapted the formula from $e^{-b}$ to $e^b$ to reflect a positive β coefficient and consequent reversal of the RAE [11] and applied this formula for those cases.

Data were standardised and Poisson regressions run in R studio using the 'glm' function of the 'stats' package [23]. $T_b$ was also added into the model in its quadratic term so we could account for the possibility of an atypical distribution of gymnasts born across the year [21].

We used the 'r.squaredLR' function from the 'MuMIn' package [24] to calculate a likelihood ratio $R^2$ in accordance with Nagelkerke [25]. Confidence intervals were calculated using the 'confint' function from the 'MASS' package [26].

## Results

Means and standard deviations, Poisson regression statistics, and the $I_D$ for each sample are outlined in Table 1. The coefficient on $T_b^2$ (our quadratic term) was nonsignificant for all our samples ($p > .05$; $R^2_{adj.}$ ranged = .00 - .13) providing no evidence of either a greater or smaller distribution of gymnasts born across the year.

### No RAE within an elite women's artistic gymnasts

There was no RAE observed within our sample of elite gymnasts competing internationally ($p = .55$; $R^2_{adj.} = .01$), a finding that remained consistent when we examined currently competing junior ($p = .14$; $R^2_{adj.} = .07$) and senior ($p = .64$; $R^2_{adj.} = .00$) gymnasts.

### RAE is conditional upon task demands

Scatter plots for the frequency of the RAE by birth week for each apparatus are shown in Fig 1.

**Beam.** A RAE favouring relatively younger gymnasts was shown in the sample of elite beam specialists ($p = .01$; $R^2_{adj.} = .12$, 95% CI [.05 - .43]). Gymnasts born at the end of the year were 27% ($I_D = 1.27$) more likely to make a World, European or Olympic beam final than those born at the start of the year.

**Vault.** Consistent with the sample of elite beam specialists, a similar RAE, favouring gymnasts born later in the year in the sample of elite, vault specialists, neared significance ($p = .08$; $R^2_{adj.} = .06$, 95% CI [-.02 - .40]). These gymnasts born at the end of the year were 21% ($I_D = 1.21$) more likely to make a World, European or Olympic vault final than those born at the start of the year.

**Uneven bars.** In comparison, to the previous two apparatus, there was no RAE ($p = .80$; $R^2_{adj.} = .00$) found in the sample of elite, uneven bars specialists.

**Floor.** A similar finding was noted for elite, floor specialists where there was no RAE within our sample ($p = .25$; $R^2_{adj.} = .03$).

**Table 1. RAE according to the Poisson regression group membership by birth week.**

| Predictor | N | $W_b$ | $T_b$ | $\beta_0$ | $\beta_1$ | $I_D$ | $R^2_{adj.}$ | 95% CI | p |
|---|---|---|---|---|---|---|---|---|---|
| International elite gymnasts | | | | | | | | | |
| Full sample | 806 | 26.25 ± 15.06 | .50 ± .29 | 2.74 | -.02 | 1.02 | .01 | [-.09, .05] | .55 |
| Juniors | 95 | 24.23 ± 14.14 | .46 ± .27 | .59 | -.15 | 1.16 | .07 | [-.36, .05] | .14 |
| Seniors | 493 | 26.87 ± 15.35 | .51 ± .30 | 2.24 | .02 | 1.02 | .00 | [-.07, .11] | .64 |
| Apparatus finalists | | | | | | | | | |
| Beam | 117 | 30.21 ± 14.37 | .57 ± .28 | .77 | .24 | 1.27 | .12 | [.05, .43] | .01*** |
| Vault | 91 | 29.29 ± 14.91 | .55 ± .29 | .54 | .19 | 1.21 | .06 | [-.02, .40] | .08* |
| Bars | 93 | 27.17 ± 15.49 | .51 ± .30 | .56 | .03 | 1.03 | .00 | [-.18, .23] | .80 |
| Floor | 105 | 28.20 ± 14.44 | .53 ± .28 | .68 | .11 | 1.12 | .03 | [-.08, .31] | .25 |

*Note.* * indicates $p < .1$,

** indicates $p < .05$,

*** indicates $p < .01$.

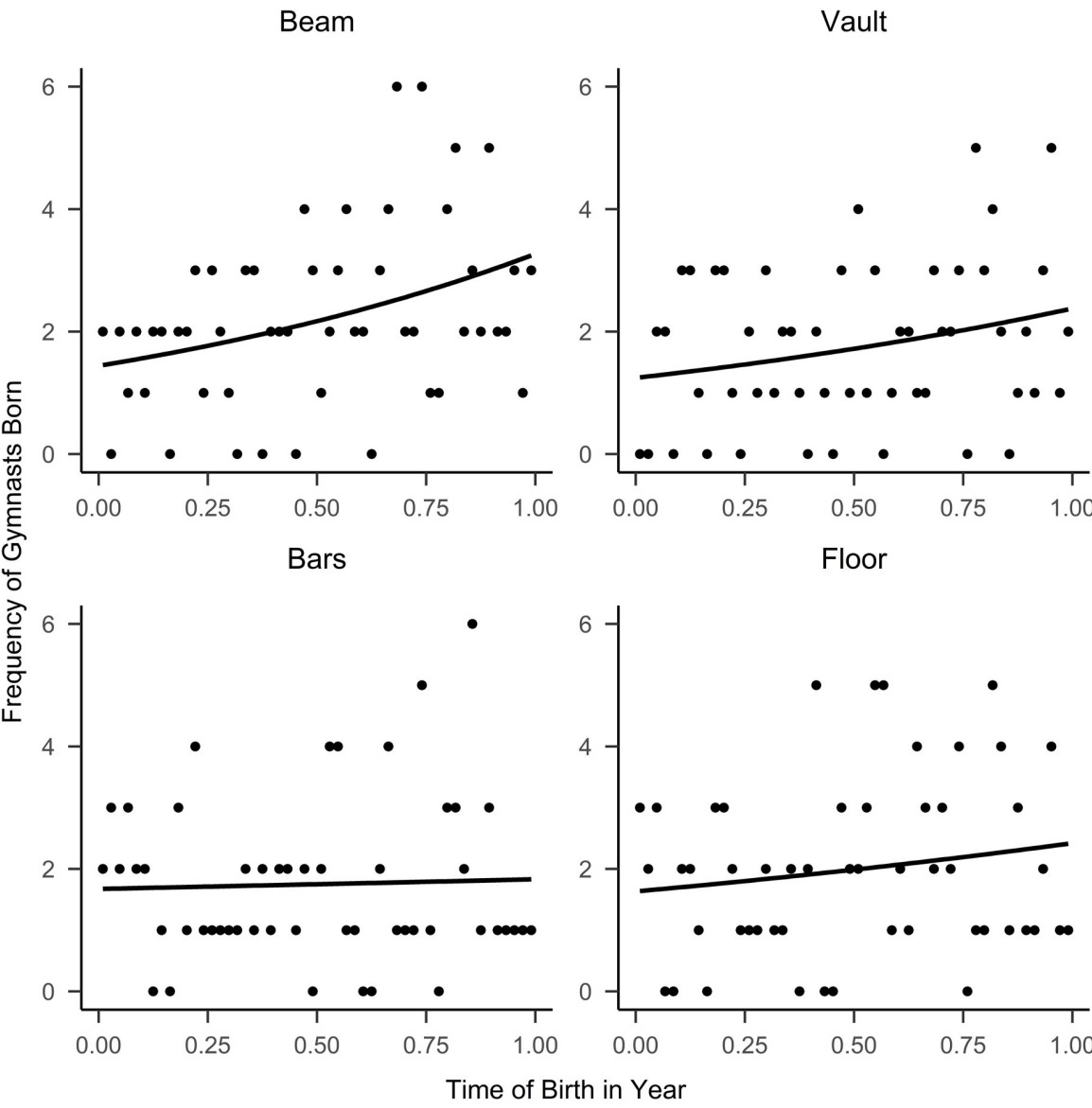

**Fig 1. Frequency of gymnasts born per week for apparatus specialists.**

## Discussion

The aim of the present research was to investigate the RAE within gymnastics, considering the influence of specific apparatus demands. In line with the previous studies exploring the RAE within women's artistic gymnastics, we hypothesised that there would be no RAE within our full sample of internationally competing gymnasts that ignored apparatus specialism. Secondly, and possibly the more novel contribution to the present literature, we hypothesised that the RAE would be conditional upon apparatus demands. Our results supported both hypotheses, revealing no RAE in the overall sample of women's artistic gymnasts that were competing at an elite, international level, but a change in the RAE when we examined the different apparatus specialisms. For gymnasts that had made a beam, and to a slightly less extent, a vault final at a major international championship (e.g., Olympics), we found that there was a greater distribution of relatively younger gymnasts in comparison to their older counterparts. Within the

sample of bars and floor specialists, however, there was an equal distribution of birth dates across the year and no evidence of a RAE.

Our expectation regarding the lack of a RAE when ignoring apparatus specialism was based on previous research in women's artistic gymnastics (e.g. Hancock et al. [11]). Similarly, Baker et al. [27] observed this "null" pattern in a sample of junior, female gymnasts and within female figure skating, another sport where athlete progression can benefit from delayed-maturation. With delayed-maturation a potential characteristic of higher-skilled gymnasts [28], a reasonable explanation for this finding is that within gymnastics, being bigger is not necessarily better and can, under certain circumstances, be detrimental. Unlike most of the RAE literature showing an overdistribution of those born earlier in the year, these findings do not support the traditionally advanced maturation hypothesis. Whilst a gymnast may not be disadvantaged by being older, the effect of being older is less dominant than in other sports; gymnasts that are relatively younger and typically smaller also possess an advantage. Even though this could indicate a bias towards these gymnasts, artistic gymnasts have been shown to grow shorter than their genetic predisposition [29] and so despite being relatively older, the advanced maturation may not be too much of a detriment. As others have theorised (e.g. Hancock et al. [11]), it is possible that previously reported null effects could be attributed to the mix of counteracting expertise levels. Cobley and colleagues (1) found that the RAE did not increase linearly with expertise, but instead the RAE at the elite level (professional / senior national representative) decreased to that of below a youth representative. We, however, controlled for this potential confound by only utilising a sample of elite, internationally competing gymnasts whilst also accounting for the age group they were competing in.

When we undertook a more subtle examination of the RAE by investigating the role of apparatus specialism, we found that gymnasts who made a beam final were 27% more likely to be born at the end of the year than born at the start. Whilst we acknowledge the potential speculation in our explanation, we feel a self-fulfilling prophecy perspective [8], likely provides the most robust explanation for these findings. Despite often being smaller, younger gymnasts are still required to develop skills at the same pace as their relatively older counterparts to enable them to be competitive. Coaches may have an expectation that these relatively younger and consequently smaller gymnasts would struggle on power events (e.g., vault). However, they may also believe that this disadvantage can be offset by a strong performance on other apparatus (e.g., beam) where size is unlikely to impact upon skill development. In turn, coaches may invest more time and resources into these younger gymnast's development on vault leading to stronger performances overall (Pygmalion effects) [10]. This theorising is reinforced by Krahenbühl and Leonardo's [30] findings which indicated that a coach's expectation of a player influenced that athletes' opportunity for participation, and resources in their sport. Support for a self-fulfilling prophecy oriented explanation of our findings is further bolstered by evidence of Galatea effects. Hancock et al. [10] explains that once expectations have been put on an individual, the individual acts in line with these expectations. With reference to our results, these gymnasts, influenced by their coaches' beliefs, could spend more time practicing on apparatus they believe that they could have success on (beam). A greater amount of deliberate practice has been consistently linked to increased performance [31] providing a complementary explanation for the increased prevalence of relatively younger gymnasts making beam finals.

Our vault findings also demonstrated an effect whereby athletes born later in the year tended to be more successful. In this instance, the challenge experienced by relatively younger athletes may enhance the development of core psychological, technical, and/or tactical skills that are needed to succeed at the highest levels [17]. Other studies providing support for the underdog hypothesis often place importance on the psychological skills (e.g., resilience, mental toughness) developed by relatively younger athletes [15]. In this case however, the implication is that the

development of superior technical skills is what provides relatively younger gymnasts with the advantage. The task demands of vaulting in gymnastics requires speed and power and the ability to "vault" over a stationary object. Due to the height of the apparatus, younger gymnasts can struggle to get over the vault as they are smaller and less powerful in comparison to their older counterparts. As these relatively younger athletes are unlikely to have maturation advantages, we theorise that coaches of these athletes will place more emphasis upon developing modifiable aspects of vaulting performance (e.g. technique). Subsequently, these gymnasts will spend more practice time ,in the developmental stages where optimum learning and motor skill development takes place [32]. This will enable such gymnasts to develop the technique needed to perform well on this apparatus and offset their potential maturation disadvantage. As gymnasts that are relatively older are typically bigger, they can rely on their height, weight, speed, and power alone to perform vaults successfully. However, as there is less apparent urgency for technical development, these gymnasts may "miss out" on developing the technical foundation needed to progress once the advantage of being bigger has disappeared. In line with Bradshaw's findings [33], having a strong technical development on vault enhances overall performance and subsequent long-term progression. This would enable the relatively younger gymnasts to undertake more difficult and challenging vaults once they reach senior levels and subsequently be more likely to make vault finals. The implications of this finding are that it is important to develop strong technical foundation, regardless of a gymnast's physical attributes. Whilst relatively older gymnasts with enhanced maturation might succeed initially, if they do not spend time refining technique, they will be less likely to excel at the higher levels.

In order to test the above theorising, future research ought to consider a younger age group of vault "specialists" to determine whether, in line with our hypotheses, there is a greater distribution of relatively older athletes. Furthermore, whilst we have a sample of junior athletes, the nature of the early specialisation sport means that most of these gymnasts are likely nearing their peak and not representative of a true developmental stage. To further our understanding of the data, it would be beneficial to use a sample that is of pre-competition age. From this, we could identify if there was an initial bias or not which would increase confidence in our theorising. It is worth noting that our results could be affected by gender bias due to the female sampling that is dominant within aesthetic sports. The magnitude of RAEs is smaller in female sports where unexpected distributions favouring Q2 athletes (athletes born in the second quarter of the year) have also been identified [34]. Further research should explore an additional sample of male gymnasts as well as investigate other gymnastics disciplines where differences in maturation and growth are prevalent [29].

There is very little research on the RAE in women's artistic gymnastics [11, 12, 27], and unfortunately, due to the nature of the samples, the conclusions drawn are limited. The samples used in the previous studies cover a time period before the notable change of scoring systems in 2006, moving away from a "perfect 10" scoring system to an open-ended system. The current Code of Points in gymnastics has brought a new level of difficulty to the sport alongside an increase in the amount of possible deductions. Because of this, research using data from before 2006 has limited implications for today. Our data, collected only after this timepoint, has superior ecological validity enabling greater confidence in the conclusions made and the relevance of our findings. Furthermore, both Baker et al. [27] and Hancock et al. [11] utilised exclusively Canadian gymnasts, most of whom competed at the provincial standard or lower [11]. Our study utilizes truly elite gymnasts from across 87 countries. Consequently, our findings have direct implications for modern-day women's artistic gymnastics and offer a genuinely global and elite perspective on the issue of the RAE.

In conclusion, our examination of intra-sport differences has added a much-needed depth, and a more sophisticated appreciation of the RAE in gymnastics. The present study is the first

to investigate apparatus specialism, utilising a contemporary analytical strategy facilitating an enhanced understanding of the theoretical underpinnings of the RAE. The findings of our study emphasise the need for RAE researchers to carefully consider both inter- and intra-sport differences for the holistic development of athletes.

## Supporting information

**S1 Table. Summary of country representation within the analysis.**
(DOCX)

## Acknowledgments

The authors would like to express thanks to Lauren Hopkins for her website containing up to date biographical information on women's artistic gymnasts. This afforded us the opportunity to collect and undertake the analysis of a large and current database.

## Author Contributions

**Conceptualization:** Eleanor Langham-Walsh, Victoria Gottwald, James Hardy.

**Data curation:** Eleanor Langham-Walsh.

**Formal analysis:** Eleanor Langham-Walsh.

**Funding acquisition:** Victoria Gottwald.

**Investigation:** Eleanor Langham-Walsh.

**Methodology:** Eleanor Langham-Walsh, Victoria Gottwald, James Hardy.

**Project administration:** Eleanor Langham-Walsh, Victoria Gottwald, James Hardy.

**Supervision:** Victoria Gottwald, James Hardy.

**Visualization:** Eleanor Langham-Walsh.

**Writing – original draft:** Eleanor Langham-Walsh.

**Writing – review & editing:** Eleanor Langham-Walsh, Victoria Gottwald, James Hardy.

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
