## [Decision Letter · Decision Letter 0]

9 Apr 2021

PONE-D-21-03611

Relative age effect? No “flipping” way! Nuanced evidence of an apparatus dependent inverse relative age effect in elite, women’s artistic gymnastics

PLOS ONE

Dear Dr. Langham-Walsh,

Thank you for submitting your manuscript to PLOS ONE. After careful consideration, we feel that it has merit but does not fully meet PLOS ONE’s publication criteria as it currently stands. Therefore, we invite you to submit a revised version of the manuscript that addresses the points raised during the review process.

Please respond to the reviewers comments in a point by point manner.

We look forward to receiving your revised manuscript.

Kind regards,

Caroline Sunderland

Academic Editor

PLOS ONE

Journal Requirements:

Reviewers' comments:

Reviewer's Responses to Questions

**Comments to the Author**

1. Is the manuscript technically sound, and do the data support the conclusions?

Reviewer #1: Yes

Reviewer #2: Partly

2. Has the statistical analysis been performed appropriately and rigorously? 

Reviewer #1: Yes

Reviewer #2: I Don't Know

3. Have the authors made all data underlying the findings in their manuscript fully available?

Reviewer #1: Yes

Reviewer #2: Yes

4. Is the manuscript presented in an intelligible fashion and written in standard English?

Reviewer #1: Yes

Reviewer #2: Yes

5. Review Comments to the Author

Reviewer #1: GENERAL REMARKS:

- Please instead R2 use R superscript 2

- Index of discrimination must be consinstently noted in manuscript (compare with line 142)

TITLE: Please be less dramatic

KEY WORDS: instead self-fulfilling prophecy – use some other key word

ABSTRACT: do not use bracet-space-text but bracet-text way of writing. Be consistent

INTRODUCTION:

- line 71 – add RAE study conducted on male olympian gymnasts (Delaš Kalinski, S., Jelaska, I., & Knežević, N. (2017). Age effects among elite male gymnasts. Acta Kinesiologica, 11(2), 84 – 89.

METHODS:

- line 136 – in formula y = e(b0+b1x) ...0 and 1 must be written in subscript and remove bracet from formula

- line 138 correct right bracet

Reviewer #2: My comments:

1) page 2, line 41, missing reference.

2) page 2 , line 43. When? Always or under which conditions? Explain RAE more thoroughly.

3) page 3, line 51-60, is or should be covered in the discussion part.

4) page 3, line 62, please explain the term inter-sport differences or rephrase the sentence.

5) page3, line 64, you make a statement of there being a bias towards delayed-maturation for success but you only seem to base that on the absence of RAE. You mention a finding of reversed RAE first in next paragraph, seems odd.

6) page 3, line 69-71, speculation, does not belong in the introduction.

7) page 4, line 75, if you don’t explain the term “Q2 gymnast” I will assume that you mean gymnast being born in the second quarter of a calendar year since this article covers the subject of relative age effect. This contradicts there being a bias toward being born late in the calendar year. What are you saying?

8) page 4, line 76, Explain “donor sport” and why this would explain the overrepresentation of gymnast being born in the second quarter in the calendar year,

9) page 4, line 90-91, What do you mean with Q1 resp Q4 effect?

10) page 5, line 98-101, Pure speculation, doesn’t fit the context.

11) page 6, line 118, maybe change “present day” to “time of writing”.

12) page 6, line 122, in my opinion, you have to include a table or some sort of description of which countries where represented and to which extent.

13) page 6, line 126, When did you obtain birth dates from Wikipedia and which one did you use (my guess is standard English Wikipedia).

14) page 6, line 130, scratch “cutting edge”.

15) page 6, line 130, just “recent”

17) page6, under Analysis just keep what describes what you did in your analysis and scratch the rest.

18) page 6, line 136, Time of birth? Time of the day??? Be more precise what you mean.

19) page 6, line 137, What competitive year?

20) page6, line 136, to me your formula doesn’t make sense. E^(b0+b1x) cant equal a birth frequency per week, have you confused frequency with rate?

21) page 7, line 149, what is middle of the year?

22) page9, line 11 and 16, no confidence interval?

23) page9, line 12 and 17, start and end of what, what is the respective definition?

24) page10, line 3-5, Belongs to introduction.

25) page 11, line 19, source? and please explain how this differs from any other sport.

26) page 12, line 11, too much speculation. 1. When did we establish that younger gymnast experience enhanced performance expectations? 2. What suggests that that younger gymnasts possess a greater psychological advantage? 3. Even if both 1 and 2 are true McKays findings doesn’t explain how they would correlate.

27) Page 12, line 13, why would the galtea effect explain that more gymnasts practicing beam or vault are born later in the calendar year? Explain.

28) Page 12 and 13, line 22-18, this is pretty much insubstantial speculation. The conclusion from more elite gymnasts being born late in calendar to that “it is important to develop a strong technical foundation, regardless of a gymnast’s physical attributes, to enable them to succeed at the higher levels”, is far-fetched. What would the implication be if it was the opposite finding with more elite level gymnast being born early?

29) page 14, line 4, Again with the Q2 athletes. What are they?

6. PLOS authors have the option to publish the peer review history of their article (what does this mean?). If published, this will include your full peer review and any attached files.

Reviewer #1: No

Reviewer #2: **Yes: **Staffan Ek

---

## [Author Response · Author response to Decision Letter 0]

17 May 2021

PLOS One: Reviewer Comments 10.05.21

Relative age effect? No “flipping” way! Inverse relative age effects in elite, women’s artistic gymnastics

Our references to page and line numbers correspond with the revised manuscript with tracked changes. 

Comments from the reviewers:

Reviewer 1:

General Remarks:

1) Please instead of R2 use R superscript 2.

Perhaps this may be a formatting issue when the document has been sent to reviewers, but we have completed a thorough search and to our knowledge, all R2 are reported using superscript. Please do highlight to us any that are not, and we will make the necessary changes. 

2) Index of discrimination must be consistently noted in manuscript (compare with line 142).

As with the above comment, to our knowledge index of discrimination is noted consistently throughout the document. Please do highlight to us any that are not, and we will edit accordingly. 

Title:

1) Please be less dramatic.

Whilst we can appreciate the reviewer’s concern over the unorthodox nature of the title, we would be keen to keep this based on how well it has been received when presenting at both national and international conferences. However, if the associate editor also feels it a little strong, we will happily change our position and amend as necessary. This comment has encouraged us to reflect further on the title, which has resulted in a more concise revised title (page 1 line 1). 

Key Words:

1) Instead self-fulfilling prophecy – use some other keyword.

We are happy to amend this keyword as long as this option is provided to us when submitting our responses to reviewer comments. We have removed self-fulfilling prophecy and included Pygmalion effects and Galatea effects instead.

Introduction:

1) Line 71 – add RAE study conducted on male Olympian gymnasts (Delaš Kalinski, S., Jelaska, I., & Knežević, N. (2017). Age effects among elite male gymnasts. Acta Kinesiologica, 11(2), 84 – 89.

Thank you for highlighting this paper to us. We have now included this within our introduction (page 4 line 74). We have also amended a sentence in our discussion to account for this (page 15 line 23). 

Methods:

1) Line 136 – in formula y = e(b0+b1x) ...0 and 1 must be written in subscript and remove bracket from formula.

Thank you we have now amended this in line with your recommendations (page 7 line 150). 

2) Line 138 correct right bracket.

This has now been amended (page 7 line 153).

Reviewer 2:

1) Page 2, line 41, missing reference.

This has now been amended (page 2 line 41).

2) Page 2, line 43. When? Always or under which conditions? Explain RAE more thoroughly.

Thank you for your comment. We have inserted additional information which explains the relative age effect more thoroughly (page 2 line 43 – page 3 line 48). We have still kept this relatively brief to avoid repetition of a later section of the introduction, discussing the influence of difference conditions upon the relative age effect (page 3 line 67 – page 5 line 109). 

3) Page 3, line 51-60, is or should be covered in the discussion part.

We felt that including a brief overview of the bio-psycho-social mechanisms underpinning the RAE, and therefore advancing beyond biological maturation alone, was important to the overall narrative of the manuscript. However, we are in agreement that this mechanistic literature also warrants addressing substantially within the discussion section. More specifically, we go on to discuss self-fulfilling prophecy in the discussion with specific reference to our results (page 13 line 4 – 23) and have incorporated further evidence supporting self-fulfilling prophecy as a mechanism for our findings (page 13 line 15 – 21). 

4) Page 3, line 62, please explain the term inter-sport differences or rephrase the sentence.

Thank you for this comment. We have provided an explanation for both inter-sport (page 3 line 67) and intra-sport (page 4 line 93) differences to enhance clarity for the reader. 

5) Page 3, line 64, you make a statement of there being a bias towards delayed maturation for success, but you only seem to base that on the absence of the RAE. You mention a finding of reversed RAE first in the next paragraph, seems odd.

Thank you for your feedback we have combined these paragraphs and made amendments to enhance the clarity, specifically identifying why there may have been a shift in the RAE across the identified age groups (page 3, line 67 – page 4 line 82) from a null distribution to a Q4 overrepresentation. We elaborate on the combined influence of delayed maturation alongside cognitive maturity. 

6) Page 3, line 67 – 71 speculation, does not belong in the introduction.

We have reworded the manuscript to enhance the clarity of where this speculation comes from (page 4, line 75).

7) Page 4, line 75, if you don’t explain the term “Q2 gymnast” I will assume that you mean gymnast being born in the second quarter of a calendar year since this article covers the subject of relative age effect. This contradicts there being a bias toward being born late in the calendar year. What are you saying?

Thank you for your comment. On reflection, we have made the decision to remove the section on Q2 gymnasts and donor sports (Page 4, line 83 - 91) as we agree that it is contradictory to other information presented in the manuscript and may lead to confusion.

8) Page 4, line 76, Explain “donor sport” and why this would explain the overrepresentation of gymnast being born in the second quarter in the calendar year.

Please see above comment. 

9) Page 4, line 90-91, What do you mean with Q1 resp Q4 effect?

We can appreciate a lack of clarity regarding the terminology used here and have subsequently elaborated on this to explain that a Q1 effect is a greater distribution of athletes born in the first quarter of the year and a Q4 effect a greater distribution of those born in the last quarter of the year (Page 5, line 100 – 103).

10) Page 5, line 98-101, Pure speculation, doesn’t fit the context.

We have removed this from the introduction (page 5 line 110 - 113). 

11) Page 6, line 118, maybe change “present day” to “time of writing”.

We have amended this in line with the above recommendation (Page 6, line 132).

12) Page 6, line 122, in my opinion, you have to include a table or some sort of description of which countries where represented and to which extent.”

We have included a table summarising the representation of each country and attached as supporting information (Page 6, line 136-137).

13) Page 6, line 126, When did you obtain birth dates from Wikipedia and which one did you use (my guess is standard English Wikipedia).

Yes, this is correct, and this has now been amended within the manuscript (Page 6, line 141).

14) Page 6, line 130, scratch “cutting edge”.

This wording has now been removed (Page 7, line 145).

15) Page 6, line 130, just “recent”.

This has now been amended (Page 7, line 145).

16) Page 6, under Analysis just keep what describes what you did in your analysis and scratch the rest.

Thank you – we have now amended this section to remove any superfluous information and just describe what we did. 

17) Page 6, line 136, Time of birth? Time of the day??? Be more precise what you mean.

We have added additional information to this to make this more precise (Page 7, line 152-154). 

18) Page 6, line 137, What competitive year?

We have amended this so that a more precise definition is now offered to explain when the year runs from (Page 7, line 152).

19) Page 6, line 136, to me your formula doesn’t make sense. E^(b0+b1x) can’t equal a birth frequency per week, have you confused frequency with rate?

We have taken this analytical strategy from Doyle and Bottomley (2018) who also use frequency of birth per week, however based upon your comments we have amended this to rate (Page 7, line 150 & 153). 

20) Page 7, line 149, what is middle of the year?

We have rephrased this as across the year to avoid ambiguity and be more in line with the continuous nature of the data (Page 8, line 166 & 174).

21) Page 9, line 11 and 16, no confidence interval?

We have amended this to include confidence intervals (Page 10, line 11 & 17).

22) Page 9, line 12 and 17, start and end of what, what is the respective definition?

We have amended this to provide the respective definitions (Page 10, line 13 & 19). 

23) Page 10, line 3-5, Belongs to introduction.

We have moved this to the introduction (Page 5, line 116-118). 

24) Page 11, line 19, source? and please explain how this differs from any other sport.

Thank you for your comment. We have chosen to delete this section from our discussion as we did not feel that it added any additional value to our findings (page 12 line 19 – page 13 line 3). 

25) Page 12, line 11, too much speculation. 1. When did we establish that younger gymnast experience enhanced performance expectations? 2. What suggests that that younger gymnasts possess a greater psychological advantage? 3. Even if both 1 and 2 are true McKays findings doesn’t explain how they would correlate.

Thank you for your comment. Although we acknowledge that this is a little speculative, a limitation of a large proportion of RAE literature, we would argue that previous research surrounding self-fulfilling prophecy and coach expectations supports our findings. To address this comment, we have added in a sentence acknowledging the speculative nature of our conclusions (page 13 line 3 & 4), whilst also including additional research supporting Pygmalion effects (page 13 line 15-18). We are also in agreement with your comment about McKay’s findings and have subsequently removed this (page 13 line 13 - 15).

26) Page 12, line 13, why would the galtea effect explain that more gymnasts practicing beam or vault are born later in the calendar year? Explain.

Thank you for your feedback. As relatively younger gymnasts are likely to be relatively smaller, it stands to reason that coaches will place an expectation that these gymnasts will not perform well on events requiring speed and power (e.g. vault). However, with most gymnasts competing on all four apparatus, this poor performance can be offset by performance on other apparatus (e.g. beam) where size is less likely to have an influence. Because of this, coaches may invest more resources into these younger gymnasts’ development on beam leading to stronger performance. Following this, the gymnast influenced by their coach’s expectations would believe that they have the potential to perform well on this event and subsequently spend more time practicing (Galatea effects). Future research may wish to pursue this further. We do not feel that Galatea effects provide an explanation for our vault findings and have instead used the underdog hypothesis as an explanation for these findings (page 14 line 6 – page 15 line 13).

We have added an additional sentence and reference to enhance the clarity of our discussion and provide a stronger explanation for why Galatea effects would result in greater practice on beam (page 13 line 19-21). Following removal of McKay’s findings from the previous paragraph, we have also combined the paragraphs examining the Pygmalion and Galatea effects which should strengthen the evidence of the link between these two effects. We have removed the discussion point regarding self-efficacy as we feel that it confused the point we were trying to make and did not add enough value to warrant its inclusion (page 13 line 24 – page 14 line 3). 

27) Page 12 and 13, line 22-18, this is pretty much insubstantial speculation. The conclusion from more elite gymnasts being born late in calendar to that “it is important to develop a strong technical foundation, regardless of a gymnast’s physical attributes, to enable them to succeed at the higher levels”, is far-fetched. What would the implication be if it was the opposite finding with more elite level gymnast being born early?

Thank you for your comment. We agree that there is a somewhat speculative nature to our conclusion as like many RAE studies. However, with research emphasising the importance of technical development for vault success (Bradshaw 2014), we think that it is important to highlight this as a possible mechanism behind our findings. We have amended the paragraph to acknowledge that our findings may be speculative (page 14 line 16), whilst also providing additional clarity for why these results may have occurred (page 14 line 16 – 19 & page 15 line 5 - 8). With regards to our implications, we have added additional depth in line the with the above amendments (page 15 line 10 – 13). 

28) Page 14, line 4, Again with the Q2 athletes. What are they?

We have amended this to provide explanation for Q2 athletes (page 15 line 24).

---

## [Decision Letter · Decision Letter 1]

10 Jun 2021

Relative age effect? No “flipping” way! Apparatus dependent inverse relative age effects in elite, women’s artistic gymnastics

PONE-D-21-03611R1

Dear Dr. Langham-Walsh,

We’re pleased to inform you that your manuscript has been judged scientifically suitable for publication and will be formally accepted for publication once it meets all outstanding technical requirements.

Kind regards,

Caroline Sunderland

Academic Editor

PLOS ONE

Additional Editor Comments (optional):

Reviewers' comments:

Reviewer's Responses to Questions

**Comments to the Author**

1. If the authors have adequately addressed your comments raised in a previous round of review and you feel that this manuscript is now acceptable for publication, you may indicate that here to bypass the “Comments to the Author” section, enter your conflict of interest statement in the “Confidential to Editor” section, and submit your "Accept" recommendation.

Reviewer #1: All comments have been addressed

Reviewer #2: All comments have been addressed

2. Is the manuscript technically sound, and do the data support the conclusions?

Reviewer #1: Yes

Reviewer #2: Yes

3. Has the statistical analysis been performed appropriately and rigorously? 

Reviewer #1: Yes

Reviewer #2: Yes

4. Have the authors made all data underlying the findings in their manuscript fully available?

Reviewer #1: Yes

Reviewer #2: Yes

5. Is the manuscript presented in an intelligible fashion and written in standard English?

Reviewer #1: Yes

Reviewer #2: Yes

6. Review Comments to the Author

Reviewer #1: All suggestions and remarks of the reviewers were adopted, and quality and clarity of the paper is further achieved. As a reviewer, I have no further objections to the new version of the paper.

Reviewer #2: The authors have made a great effort to improve the article with the comments that the reviewers have presented.

7. PLOS authors have the option to publish the peer review history of their article (what does this mean?). If published, this will include your full peer review and any attached files.

Reviewer #1: **Yes: **Sunčica Delaš Kalinski

Reviewer #2: **Yes: **Staffan Ek

---

## [Editor Report · Acceptance letter]

18 Jun 2021

PONE-D-21-03611R1 

Relative age effect? No “flipping” way! Apparatus dependent inverse relative age effects in elite, women’s artistic gymnastics 

Dear Dr. Langham-Walsh:

I'm pleased to inform you that your manuscript has been deemed suitable for publication in PLOS ONE. Congratulations! Your manuscript is now with our production department. 

Kind regards, 

on behalf of

Dr. Caroline Sunderland 

Academic Editor

PLOS ONE